# Intestinal SIRT1 Deficiency-Related Intestinal Inflammation and Dysbiosis Aggravate TNFα-Mediated Renal Dysfunction in Cirrhotic Ascitic Mice

**DOI:** 10.3390/ijms22031233

**Published:** 2021-01-27

**Authors:** Yu-Te Chou, Tze-Tze Liu, Ueng-Cheng Yang, Chia-Chang Huang, Chih-Wei Liu, Shiang-Fen Huang, Tzu-Hao Li, Hsuan-Miao Liu, Ming-Wei Lin, Ying-Ying Yang, Tzung-Yan Lee, Yi-Hsiang Huang, Ming-Chih Hou, Han-Chieh Lin

**Affiliations:** 1Department of Medicine, Taipei Veterans General Hospital, Taipei 11267, Taiwan; ytchou5@vghtpe.gov.tw (Y.-T.C.); cwliu2@vghtpe.gov.tw (C.-W.L.); yhhuang@vghtpe.gov.tw (Y.-H.H.); mchou@vghtpe.gov.tw (M.-C.H.); 2Faculty of Medicine, School of Medicine, National Yang Ming Chiao Tung University, Taipei 11267, Taiwan; ttliu@ym.edu.tw (T.-T.L.); yang@ym.edu.tw (U.-C.Y.); cchuang7@vghtpe.gov.tw (C.-C.H.); sfhuang6@vghtpe.gov.tw (S.-F.H.); thli3@ym.edu.tw (T.-H.L.); mwlin@ssssym.edu.tw (M.-W.L.); 3Genomic Research Center, National Yang Ming Chiao Tung University, Taipei 11267, Taiwan; 4Institute of Biomedical Informatics, Taipei 11267, Taiwan; 5Division of Clinical Skills Training Center, Department of Medical Education, Taipei Veterans General Hospital, Taipei 112, Taiwan; 6Institute of Clinical Medicine, National Yang-Ming University, Taipei 11267, Taiwan; 7Division of Infection, Taipei Veterans General Hospital, Taipei 11267, Taiwan; 8Division of Allergy, Immunology, and Rheumatology, Department of Internal Medicine, Shin Kong Wu Ho-Su Memorial Hospital, Taipei 11267, Taiwan; 9Graduate Institute of Traditional Chinese Medicine, Chang Guang Memorial Hospital, Linkou 33371, Taiwan; miaowhale@gmail.com (H.-M.L.); joyamen@mail.cgu.edu.tw (T.-Y.L.); 10Institute of Public Health, National Yang-Ming University, Taipei 11267, Taiwan; 11Division of Gastroenterology and Hepatology, Taipei Veterans General Hospital, Taipei 11267, Taiwan

**Keywords:** cirrhotic ascites, tumor necrosis factor-α (TNFα), NAD-dependent deacetylase sirtuin-1 (SIRT1), intestinal barrier dysfunction, intestinal dysbiosis, renal dysfunction

## Abstract

In advanced cirrhosis, the TNFα-mediated intestinal inflammation and bacteria dysbiosis are involved in the development of inflammation and vasoconstriction-related renal dysfunction. In colitis and acute kidney injury models, activation of SIRT1 attenuates the TNFα-mediated intestinal and renal abnormalities. This study explores the impacts of intestinal SIRT1 deficiency and TNFα-mediated intestinal abnormalities on the development of cirrhosis-related renal dysfunction. Systemic and renal hemodynamics, intestinal dysbiosis [cirrhosis dysbiosis ratio (CDR) as marker of dysbiosis], and direct renal vasoconstrictive response (renal vascular resistance (RVR) and glomerular filtration rate (GFR)) to cumulative doses of TNFα were measured in bile duct ligated (BDL)-cirrhotic ascitic mice. In SIRT1*^IEC-KO^*-BDL-ascitic mice, the worsening of intestinal dysbiosis exacerbates intestinal inflammation/barrier dysfunction, the upregulation of the expressions of intestinal/renal TNFα-related pathogenic signals, higher TNFα-induced increase in RVR, and decrease in GFR in perfused kidney. In intestinal SIRT1 knockout groups, the positive correlations were identified between intestinal SIRT1 activity and CDR. Particularly, the negative correlations were identified between CDR and RVR, with the positive correlation between CDR and GFR. In mice with advanced cirrhosis, the expression of intestinal SIRT1 is involved in the linkage between intestinal dysbiosis and vasoconstriction/hypoperfusion-related renal dysfunction through the crosstalk between intestinal/renal TNFα-related pathogenic inflammatory signals.

## 1. Introduction

Renal failure is a challenge in cirrhotic patients because the likelihood of mortality occurring increases with worsening renal function [1]. One of the most severe forms of renal dysfunction in advanced cirrhosis is hepatorenal syndrome (HRS), which is accompanied by systemic, intestinal, renal inflammation as well as renal vasoconstriction [2,3,4,5]. At present, the search for new potential agents to treat patients with cirrhotic HRS whose responses to standard treatments are poor is ongoing [6,7].

In advanced cirrhosis, persistent increased circulating tumor necrosis factor-α (TNFα)-related bacterial translocation and systemic (hepatic, intestinal, renal) inflammation are involved in the development of intestinal/renal dysfunction and HRS [3,5]. In cirrhotic patients at diagnosis and resolution of the infection, the development of renal impairment is associated with significantly high plasma and ascitic fluid TNFα levels [8]. Agents with anti-TNFα effects can prevent the development of systemic inflammation and renal dysfunction in cirrhotic rats with portal hypertension [9,10].

In healthy rats, the infusion of high doses of TNFα lowers blood pressure, glomerular filtration rate and renal blood flow as well as directly induces renal vasoconstriction and increases renal vascular resistance [11,12]. Advanced cirrhotic patients with HRS are characterized by marked renal vasoconstriction, reduced renal blood flow and perfusion [2,4]. Increased circulating TNFα is involved in the pathogenesis of HRS, and higher plasma TNFα levels were observed in cirrhotic patients with HRS than in those without HRS [2,3,9,10,13]. TNFα inhibitors, such as pentoxifylline, can prevent and improve HRS in advanced cirrhotic patients [9,10].

In cirrhotic rats with ascites, anti-TNFα monoclonal antibody administration directly suppresses intestinal inflammation, reduces intestinal barrier dysfunction, and decreases the incidence of bacterial translocation [14,15]. Increased renal TNFα expression indicates the priming of cirrhotic kidneys by chronic intestinal inflammation, barrier dysfunction, and bacterial translocation [5,16]. Chronic intestinal decontamination with rifaximin, non-absorbed gut-directed antibiotic, decreases the severity of acute kidney injury and HRS by reducing serum TNFα levels in patients with advanced cirrhosis [16]. However, side effects, such as nausea, vomiting, flatulence and abdominal pain, high cost, resistance, and the hepatotoxicity of anti-TNFα antibody limit the application of agents with anti-TNF effects in advanced cirrhotic patients with circulatory and renal dysfunction. Therefore, for cirrhotic patients with HRS, the identification of the upstream origin of systemic and local TNFα, to simultaneously treat intestinal inflammation, intestinal barrier dysfunction, intestinal dysbiosis, bacterial translocation, systemic inflammation and renal dysfunction is urgently needed.

In rats with DDS-induced colitis, SIRT1 activator pretreatment corrects colonic dysbiosis and reduces systemic and colonic mucosa inflammation [17]. SIRT1 *RNA* and protein expression are reduced in whole intestinal biopsies and the lamina propria mononuclear cells of patients with inflammatory bowel disease (IBD) [18]. In patients with IBD, treatment with infliximab, a chimeric monoclonal antibody against TNFα, restores the intestinal mucosal expression of SIRT1 [18]. Mice with intestinal deletion of SIRT1 (SIRT1*^IEC-KO^*) had abnormal activation of the TNFα pathway, gut dysbiosis and severe intestinal inflammation [19]. Pharmacological activation of intestinal SIRT1 attenuates TNFα-mediated intestinal barrier dysfunction, inflammation, and dysbiosis [20,21]. In *Toxoplasma gondii*-infected mice, pretreatment with SIRT1 activator significantly suppresses TNFα levels in the ileum, mesenteric lymph nodes and spleen, preventing intestinal barrier dysfunction, reducing bacterial translocation and improving intestinal dysbiosis [21].

A recent study reported an increased risk of chronic kidney disease (CKD) in cases with inflammatory bowel disease [22]. In CKD, the suppression of intestinal bacteria dysbiosis significantly attenuates the severity of renal dysfunction [23]. In acute kidney injury model, anti-TNFα, antioxidant activities and acute pharmacologic activation of SIRT1 induces reno-vasodilatation, increases renal blood flow (RBF) and decreases RVR [24,25,26]. In diabetic nephropathy animals, chronic pharmacological activation of SIRT1 improved renal dysfunction by decreasing plasma and renal TNFα [27]. Systemic sirtuin 1 (SIRT1) activation reduces portal pressure by downregulating hepatic TNFα expression, inhibiting hepatic inflammation, and suppressing intrahepatic vasoconstriction [28]. Nonetheless, the contribution of the crosstalk between intestinal and renal TNF and SIRT1 signals in the severe renal dysfunction of cirrhosis has not yet been explored.

Accordingly, this study comprehensively explores the impacts of intestinal SIRT1 deficiency on TNFα-mediated intestinal inflammation, intestinal barrier dysfunction, intestinal bacterial dysbiosis, bacterial translocation on the development of severe renal dysfunction in cirrhotic mice with ascites.

## 2. Results

### 2.1. Intestinal SIRT1 Deficiency Aggravates Severity of Renal Dysfunction in SIRT1*^IEC-KO^*-BDL-Cirrhotic Mice with Ascites

Compared to WT-sham mice, the WT-BDL mice showed typical cirrhotic livers, ascites, upregulated hepatic TNFα-TNFR1 signals, significant circulatory dysfunction (reduced MAP, decreased CO and CI), elevated serum ALT, bilirubin and creatinine levels and hyponatremia, decreased urine output and decreased body weight (Figure 1 and Table 1). In SIRT1*^IEC-KO^* BDL-cirrhotic mice with ascites, the deletion of intestinal SIRT1 expression (Figure 1B and Table 1) aggravated the abovementioned abnormalities observed in WT-BDL mice (Figure 1 and Table 1). However, the serum ALT level and hepatic TNFR2 signals were not different between WT-BDL and SIRT1*^IEC-KO^* BDL mice. Notably, the greater increase in jaundice was not associated with a greater increase in serum ALT level in SIRT1*^IEC-KO^*-BDL mice compared to those in WT-BDL mice. The heart rate was not different among the four groups (Table 1).

### 2.2. Genetic Deletion of Intestinal SIRT1 Exacerbates TNFα-Mediated Intestinal Inflammation and Barrier Dysfunction in SIRT1*^IEC KO^*—BDL Mice

In WT-BDL mice with a higher intestinal TNFα level, higher ileal mucosal injury score, more fecal albumin loss, and worsening Evans blue-assessed barrier dysfunction were observed compared to WT-sham mice. Additionally, the upregulation of intestinal lipocalin-2, TNFR1, TNFR2, p-MLCK was associated with the downregulation of intestinal SIRT1, catalase, CuZnSOD, MnSOD, occludin, ZO-1 and E-cadherin expressions in WT-BDL mice (Figure 2). Compared to WT-BDL mice, intestinal deletion of SIRT1 aggravates the abovementioned intestinal abnormalities in SIRT1*^IEC KO^*-BDL-cirrhotic mice (Figure 2). Nonetheless, the expression levels of intestinal TNFR2 and E-cadherin were not different between WT-BDL and SIRT1*^IEC KO^*-BDL mice (Figure 2E).

### 2.3. Intestinal SIRT1 Deficiency Aggravated Renal Dysfunction through the Upregulation of TNFα-Activated Signals in SIRT1*^IEC KO^*-BDL Mice

Compared to WT-sham mice, WT-BDL mice with severe renal dysfunction had higher levels of urinary KIM-1 and NGAL, higher renal tubular injury score, tubular interstitial fibrosis score, renal TNFα and TBARS levels, expressions of renal inflammatory (TNFR1 and TNFR2), and injury (NOX-2, KIM-1 and lipocalin-2) markers as well as lower renal expressions of SIRT1, anti-oxidants (catalase, CuZnSOD and MnSOD) and barrier (ZO-1, occludin, E-cadherin) markers (Figure 3). In SIRT1*^IEC KO^*-BDL mice, the deletion of intestinal SIRT1 expression (Figure 1B and Figure 2, and Table 1) further aggravates the abovementioned renal abnormalities of WT-BDL mice (Figure 3).

### 2.4. Direct Evidence for the Aggravation of TNFa-Induced Renal Dysfunction in SIRT1*^IEC KO^* BDL-Cirrhotic Mice due to the Deficiency of Intestinal SIRT1

In the in situ renal perfusion study of WT-BDL mice, the lower basal MAP, GFR, RBF and the higher RVR were noted compared to those of WT-sham mice (Figure 4). Notably, the magnitude (percentage change) of the TNFα-induced decrease in MAP was not different between the four groups (WT-BDL-cirrhotic and SIRT1^IEC-KO^-BDL-cirrhotic mice) (Figure 4A). A higher degree of TNFα-induced increase in the RVR and the decrease in GFR and RBF were observed among SIRT1*^IEC KO^*-BDL-cirrhotic ascitic mice than in WT-BDL-cirrhotic ascitic mice (Figure 4B–D). However, the degree of TNFα-induced changes in GFR, RVR and RBF did not differ between WT-sham and SIRT1*^IEC KO^*-sham mice.

### 2.5. Genetic Deletion of Intestinal SIRT1 Aggravates the Intestinal Bacterial Dysbiosis in SIRT1*^IEC KO^*-Mice

The Shannon and *Faith’s* Phylogenetic Diversity (PD) index, and evenness index were significantly lower in the WT-BDL mice and SIRT1*^IEC KO^*-BDL mice than in the WT-sham mice (Figure 5A and Appendix A). These results indicate that the overall microbial species diversity was lower in the WT-BDL mice and the SIRT1*^IEC KO^*-BDL mice than in the WT-sham mice. The UniFrac principal coordinate analysis (PCoA) is shown in Figure 5B. The WT-sham and WT-BDL groups were separated into different clusters (PERMANOVA, *p* = 0.001, Q valve = 0.006). Likewise, the microbiota of the SIRT1*^IEC KO^*-BDL group was clustered separately from that of WT BDL group (PERMANOVA, PERMANOVA, *p* = 0.003, Q valve = 0.008).

Notably, the Taxa bar plot (Appendix A) of the 10 most abundant taxa shows the same trends as the chord diagram. The chord diagram indicates that the intestinal microbiota of the WT-BDL mice at the phylum level were characterized by a decrease in anti-inflammatory *Firmicutes* bacteria and an increase in proinflammatory *Deferribacteres* and *Epsilobacteraeota* bacteria compared to those in WT-sham mice (Figure 5C). In SIRT1*^IEC KO^*-BDL mice, an increase in proinflammatory *Bacteroidetes* bacteria compared to in WT-BDL mice was observed. The decreased in abundance of *Firmicutes* was associated with an increase in proinflammatory *Proteobacteria* bacteria in SIRT1*^IEC KO^*-sham mice.

At the class and order levels, the intestinal microbiota of WT-BDL mice were characterized by a decrease in the abundance of anti-inflammatory bacteria *Clostridia* and *Clostridiales* (belong to *Firmicutes* phylum) and the increased proinflammatory bacteria *Campylobacteria*, *Deferribacteres, Defferibacteres* (belong to *Deferribacteres* phylum) compared WT-sham mice (Appendix A). Compared to WT-BDL mice, a further increase in the abundance of *Bacteroidia,*
*Bacteroidales* (belong to *Bacteroidetes* phylum) and *Defferibacterales* was observed in SIRT1*^IEC-KO^*-BDL mice. Notably, the presence of proinflammatory *Gammaproteobacteria and Enterobacteriales* (belong to *proteobacteria* phylum) bacteria in intestines of SIRT1*^IEC KO^*-sham mice was observed (Appendix A). At the family and genus levels, in comparison with WT-sham mice feces, the absence of anti-inflammatory bacteria [*Lactbacillaceae* (family)/*Lactobacillus* (genus)] was associated with increase in proinflammatory bacteria *(Muribaculaceae* (family, belong to *Deferribacteres* phylum), *Mucispinillum* (genus, belong to *Deferribacteres* phylum), *Helicobaraceae*) in WT-BDL mice feces (Appendix A). Notably, SIRT1*^IEC KO^*-BDL mice were characterized by the presence of proinflammatory bacteria (*Peptostrepococcaceae* (family, belong to *Clostridiaes* order), *Bacteroidaceae* (family), *Romboutsia* (genus, belong to *Peptostreptococcaceae* family), *Bacteroides* (genus), and *Parabacteroides* (genus), *Muribaculaceae* (family, belong to *Deferribacteres* phylum)) compared to the lack of these in WT-BDL mice feces (Appendix A). In comparison with WT-sham mice, SIRT1*^IEC KO^*-sham mice were characterized by higher levels of intestinal *Enterbacteriaceae* and lower levels of *Lactbacillaceae* at the family level; as well as the presence of *Escherichia*-*shigella*, and *Parabacteroides,* and *Citrobacter* at genus level (Appendix A).

### 2.6. Shifting to Proinflammatory Bacteria that have Biological Relevance for Intestinal Epithelial Barrier Dysfunction in SIRT1*^IEC-KO^*-BDL Mice

In WT-sham mice, both the cladogram (Appendix A) and the linear discriminant analysis (LDA) effect size method (LAD score > 3; Figure 6 and Figure 7) revealed an increase in beneficial bacteria (*Clostridia, Clostridiales, Lachnospiraceae, Lactobacillale, Lactobacillaceae* (belong to *Defirmicutes*), short chain fatty acid producer *Prevotellaceae* (belong to order *bacteroidales*), *Ruminococcaceae* UCG and *Ruminococcus*1 and *Ruminiclostridium* 6) that have involvement in the integrity of intestinal epithelial barrier. In contrast, the intestinal microbiota of WT-BDL mice were characterized by an increase in pathogenic bacteria (*Rikenellaceae* (belong to *Bacteroidetes* rhylum), *Alistipes* (belong to *Rikenellaceae* family), *Parasutterella* (belong to *Proteobacteria* phylum), *Defferibacterales*, *Deferribacteres*, *Deferribacteraceae*, and *Mucispirillum* (belong to *Deferribacteres* phylum) that compromise barrier function (Figure 6 and Figure 7).

Notably, SIRT1*^IEC KO^*-BDL mice were characterized by an increase in intestinal inflammatory bacteria (*Bacteroides*, *Bacteroidaceae*, *Peptostreptococcaceae* (belong to *Clostridiales* order), *Romboutsia* (belong to family of *Peptostreptococcaceae*), *Ruminococcaceae*, *Ruminiclostridium* 6, *Blautai* and *Ruminiclostridium* (belong to *Clostridia* class), *Delatproteobacteria*, *Desulfovibrio, Desulfavibrionales* (belong to *Deltaproteobacteria* phylum), *Streptococcaceae* (belong to *Bacilli* class), *Streptococcus* (belong to *Streptococcaceae* family and *Bacilli* class), *Actinobacteria*, *Eggerthellaceae*, *C**oriobacteriales, C**oriobacteria* (belong to *Actinobacteria* phylum) that compromise the barrier function (Figure 6 and Figure 7).

In addition to preserving anti-inflammatory bacteria, including Verrucomicrobiales, Verrucomicrobia, Verrucomicrobiales, Akkermansia, Akkermansiaceae, Erysipelptrichales, Erysipelotrichaceae, Enterococcus, Enterococcaceae (belong to Firmicutes rhylum), the microbiota of SIRT1^IEC KO^-sham mice were characterized by an increased abundance of proinflammatory bacteria including Proteobacteria, Gammaproteobacteria, Betaproteobacteriales, Enterobacteriaceae, Delftia, Escherichia-shigella (belong to Proteobacteria phylum), Patescibacteria, Saccharimonadia, Saccharimonadates, Saccharimonadaceae (belong to Patescibacteria phylum), which compromise the barrier function (Figure 6 and Figure 7).

### 2.7. Intestinal Bacterial Dysbiosis was Associated with the Upregulation of Renal Inflammatory and Barrier-Disrupted Markers in SIRT1*^IEC-KO^* Mice

Compared to WT groups, a significant positive correlation was noted between *mRNA*/protein levels of renal inflammatory mediators (lipocalin-2 and TNFR1) and the abundance of barrier-disrupted pathogenic intestinal bacteria (*Epsilonbacteraeota* (phylum) and *Campylobacteria* (class)) in KO groups (Figure 8A,B). In comparison with WT groups, the negative correlation between *mRNA*/protein levels of renal inflammatory mediators (lipocalin-2 and TNFR1) and the abundance of barrier-protected intestinal bacteria (*Bacilli* (class), *Saccharimonadia* (class), *Erysipelotrichia* (class), *Verrucomicrobia* (class) was noted in KO groups. Significantly, there was positive correlation between the *mRNA*/protein levels of anti-inflammatory (catalase, MnSOD and CuZnSOD)/barrier (occludin and E-adhesin) markers and the abundance of barrier-protected intestinal bacteria (*Bacilli* (class), *Erysipelotrichia* (class), *Verrucomicrobia* (class)). Additionally, there was a negative correlation between the *mRNA*/protein levels of anti-inflammatory (catalase, MnSOD, and CuZnSOD)/barrier (occludin and E-adhesin) markers and the abundance of barrier-disrupted intestinal bacteria (*Saccharimonadia* (class), *Episolbacteraeata*, *Campylobacteria*) in KO groups.

This correlation analysis was performed to explore the crosstalk between the abundance of intestinal bacteria (with inflammatory, anti-inflammatory, barrier-protected and barrier-disrupted effects), protein/*mRNA* levels of intestinal/renal injured, anti-inflammatory, barrier-protected, and barrier-disrupted markers. Notably, the trend of the correlation between the abundance of intestinal microbiota across phylum, class and order levels, and the levels of intestinal pathogenic/protective markers (Appendix A) were similar to those between intestinal microbiota and renal pathogenic/protective markers (Figure 9A,B). Moreover, the significant correlations were noted between the levels of intestinal and renal protective (ZO-1, occludin, E-adhesin, catalase, MnSOD, and CuZn-SOD) as well as pathogenic markers (TNFR1, TNFR2, MLKC, p-MLKC, lipocalin-2 and KIM-1) (Figure 9A,B).

### 2.8. Direct Link between Intestinal Bacterial Dysbiosis and Severe Renal Dysfunction in SIRT1*^IEC-KO^*-BDL Mice

Compared to WT-sham mice, higher serum LPS binding protein (LBP) levels (Table 1), higher MLN culture positive rates (Table 1), and lower cirrhosis dysbiosis ratios (CDR) (Figure 10A) were observed in WT-BDL mice. Notably, the deletion of the intestinal SIRT1 gene aggravated the degree of increase in serum LBP level and MLN culture positive rate, as well as the decrease in intestinal CDR in SIRT1*^IEC-KO^*-BDL mice compared to those in WT-BDL mice. Figure 10B shows a positive correlation between data of the intestinal CDR and SIRT1 activity of WT and KO mice. The correlation was stronger between KO than WT groups, and significance was only noted between KO groups. Significantly, there was a negative correlation between the level of intestinal SIRT1 activity and baseline RVR in KO groups (Figure 10C). In the KO groups, the significant negative correlation between data of the intestinal CDR and the data of RVR was parallel to the positive correlation between the data of intestinal CDR and the data of the GFR (Figure 10D,F). In both the WT and KO groups, the correlation between intestinal SIRT1 activity and the data of GFR did not reach statistical significance (Figure 10E).

## 3. Discussion

Pharmacologic and genetic depletion of systemic TNFα can suppress aging-associated intestinal and systemic inflammation and normalize intestinal dysbiosis [19]. The preservation of the gut microbiota in older TNF^−/−^ mice indicates the important pathogenic roles of TNFα in age-related intestinal dysbiosis, systemic inflammation and intestinal barrier dysfunction [29]. Increased abundance of *Rikenellaceae* family is associated with the exacerbation of intestinal inflammation [30]. In cirrhotic patients, intestinal inflammation and barrier dysfunction, overgrowth of the proinflammatory intestinal bacteria, such as *Enterobacteriaceae*, *Veillonellaceae*, and *Streptococcaceae)* and decreased anti-inflammatory *Lachnospiraceae* bacteria abundance are correlated with the severity of cirrhosis [29,30,31,32]. The exogenous administration of *Akkermansia* attenuates metabolic endotoxemia-induced inflammation through restoration of the gut barrier [33]. In this study, the anti-inflammatory intestinal bacteria, including *Lachnospiraceae* and *Akkermansia*, were dominant in sham mice, whereas the proinflammatory intestinal bacteria, including *Streptococcaceae* and *Rikenellaceae*, were dominant in WT-BDL and SIRT1*^IEC KO^*-BDL mice with severe intestinal barrier dysfunction. The cirrhosis dysbiosis ratio (CDR), which represents the balance between proinflammatory and anti-inflammatory intestinal bacteria, is important for prediction of cirrhosis-related intestinal and renal complications [31,32].

In particular, this study revealed a significant positive correlation between the CDR and intestinal SIRT1 activity, between the CDR and glomerular filtration rate, as well as a negative correlation between CDR and renal vascular resistance. This emphasizes the involvement of intestinal proinflammatory and anti-inflammatory bacteria in the intestinal and renal dysregulation of cirrhosis (Appendix A). TNFα causes intestinal tight junction disruption and subsequent intestinal barrier dysfunction via myosin light chain kinase (MLCK) activation [34,35]. The upregulation of intestinal TNFα-TNFR signals and phosphorylation of intestinal MLCK is involved in the pathogenesis of intestinal barrier dysfunction and disease progression in cirrhosis [36]. In SIRT1*^IEC-KO^*-BDL-cirrhotic mice, the depleted intestinal SIRT1-related upregulation of TNFα-TNFR signals are associated with the upregulation of intestinal p-MLCK expression and disruption of intestinal barrier integrity. In the cirrhotic mice, there is a positive correlation between the expression levels of intestinal p-MLCK and renal lipocalin-2 (inflammatory and renal injury marker) as well as intestinal p-MLCK and renal KIM-1.

Upon inflammation and tissue damage, in addition to crosstalk in the intestine, activation of renal SIRT1 attenuates the detrimental effects of TNFα on renal mesangial and tubular epithelial cells [19,20,21,24,25,26,27]. The deficiency of intestinal SIRT1 exacerbates TNFα-mediated renal damage in mice with cholestasis [37]. In a nephropathy model, systemic SIRT1 activation suppresses intestinal/renal TNFα expression and ameliorates renal dysfunction [27]. In this study, the elevation of serum bilirubin was associated with increased creatinine level in SIRT1*^IEC-KO^*-BDL-cirrhotic mice compared to WT BDL-cirrhotic mice (Figure 1E,F). In addition to supporting the previous observation regarding the crosstalk between hepatic SIRT1 and TNFα [28], this study suggests a link between SIRT1−TNFα crosstalk and intestinal SIRT1-related effects and renal dysfunction of cirrhosis. 

In response to inflammation and tissue injury, TNFα directly induces the production of lipocalin-2 from intestinal and renal epithelial cells [38]. Lipocalin-2 is an inflammatory mediator for intestinal and renal inflammation and is positively correlated with inflammatory disease severity [38,39,40]. Serum lipocalin-2 are positively correlated with serum levels of soluble TNFR and negatively correlated with glomerular filtration rate [40]. Renal expression of lipocalin-2 is significantly increased in decompensated cirrhosis with increased circulating TNFα and acute renal injury [40,41]. In this study, there was a positive correlation between the abundance of proinflammatory bacteria and intestinal/renal lipocalin/TNFα/TNFR expression in the WT and KO groups (Appendix A). Accordingly, these observations reinforced the existence of inflamed and leaky intestine-driven renal dysfunction in SIRT1*^IEC-KO^*-BDL-cirrhotic mice.

Chronic kidney disease (CKD) is characterized by the accumulation of metabolites of proinflammatory gut bacteria. These can damage renal tubular cells by increasing cellular oxidative stress and deteriorating renal function [42,43]. In CKD, the suppression of intestinal bacteria dysbiosis by the consumption of a high-fiber diet significantly attenuates the disruption of the colonic epithelial tight junction and severity of renal dysfunction [23]. In patents with CKD, there is an increased abundance of proinflammatory *enterobacteriaceae* and *Clostridiaceae* bacteria and decreased abundance of anti-inflammatory butyrate-producing bacteria (*Lactobacillaceae* and *Prevotellaceae*) [43]. In the present study, intestinal inflammation and renal dysfunction in WT-BDL and SIRT1*^IEC-KO^* BDL-cirrhotic mice were characterized by increased intestinal abundance of *Enterobacteriaceae* and *Clostridiaceae* as well as decreased abundance of *Lactobacillaceae* and *Prevotellaceae*.

In the model of murine colitis, *TNFR1* is highly expressed in intestinal epithelial cells and mediates TNFα induced inflammatory cascades [42,44]. In the present study, the SIRT1-related modulation of intestine TNFα expression primarily occurs through the upregulation of TNFR1 expression in the cirrhotic intestine. Meanwhile, TNFR1 is highly expressed in the renal proximal tubule and collecting duct, TNFR1 activation reduces renal blood flow (RBF) and glomerular filtration rate (GFR) [45,46]. In patients with diabetic kidney disease, serum TNFR levels are positively associated with estimated glomerular filtration rate (eGFR) decline and disease severity [47,48]. In SIRT1*^IEC-KO^* BDL-cirrhotic mice the detrimental renal effects of intestinal SIRT1 deficiency, including further decrease in RBF and GFR, occurred through the simultaneous upregulation of intestinal and renal TNFα-TNFR1 cascades (Appendix A). Additionally, there were significant positive correlations between oxidative stress markers (intestinal and renal MnSOD, CuZn-SOD, catalase), inflammation marker (intestinal and renal TNFR1/TNFR2), and intestinal/renal dysfunction markers (p-MLCK, MLCK, lipocalin-2 and KIM).

Resveratrol has rapid metabolism and low bioavailability. After resveratrol was ingested 77–80% of it was absorbed in the intestine and 49–60% of this is excreted in the urine. After ingestion resveratrol also reaches the intestine via the hepatic portal system. Decreased intestinal SIRT1 had been associated with intestinal and renal inflammation [18,19,24]. Anti-TNFα-related anti-inflammatory effects of SIRT1 activator resveratrol had been reported in chondrogenic mesenchymal stem and tumor colon rectal cancer cell system [49,50,51]. In experimental models of acute and chronic kidney injury, chronic systemic SIRT1 activation with resveratrol significantly improves adriamycin-induced, subtotal nephrectomy-induced and unilateral ureteral obstruction-induced renal dysfunction through the anti-inflammatory effects [24,52,53]. Through SIRT1 activation, resveratrol treatment attenuated the intestinal inflammation and renal dysfunction in the noncirrhosis model [17,20,21,24,49,50,51,52,53].

## 4. Conclusions

This study reinforced the concepts of increased TNFα-mediated renal oxidative stress and inflammation as the main culprits for the deterioration of renal function in cirrhotic patients. In cirrhotic ascitic mice, the well-expressed intestinal SIRT1 is crucial to prevent TNFα-mediated activation of systemic, intestinal and renal oxidative stress, inflammation and injured signals that contribute to the development of renal dysfunction. Agents that can restore intestinal SIRT1 have the potential to improve the inflammation-derived TNFα-mediated renal dysfunction in advanced cirrhosis. So, it is noteworthy to explore the effects of the administration of a highly absorbed intestinal SIRT1 activator such as resveratrol on the coexisting TNFα-mediated intestinal and renal pathogenic changes as well as renal dysfunction in cirrhosis.

## 5. Materials and Methods

### 5.1. Animals

Animals used in this study were 10 weeks-old male mice with an intestinal epithelial specific deletion (knockout, KO) of the SIRT1 gene (SIRT1*^IEC-KO^*) with a 99% C57BL/6J genetic background. To generate SIRT1*^IEC-KO^* mice, SIRT1 flox/flox mice were crossed with transgenic mice expressing Cre recombinase under the control of the villin promoter (VillinCre−), which is expressed in intestinal epithelial cells (IEC) purchased from Charles River Japan, Inc. (Yokohama, Japan). Wild type C57BL/6 mice were served as controls (WT). The experiments were approved by the animal ethical committee of Yang-Ming medical university with approval of No. 1061232r and IACUC 2018-54. All the raising and breeding of animals is undertaken in the laboratory animal center of National Yang-Ming University. All the experiments were performed in our laboratory. All efforts were made to minimize animal suffering by administering inhalation anesthetics (isoflurane). At the end of the experiments, the mice were euthanized with the inhalation anesthetic by overdose of isoflurane.

### 5.2. Groupings

Common bile duct ligation (BDL) was undergone on ten week-old WT and SIRT1*^IEC-KO^* mice to create severe renal dysfunction in advanced cirrhosis [54]. Six weeks after BDL, ≅15% of BDL mice died. The presence of ascites was evaluated by ultrasound and 85% of BDL mice displayed ascites at the time of sacrifice. This created 4 experimental groups namely WT sham, WT-BDL, SIRT1*^IEC-KO^*-sham, and SIRT1*^IEC-KO^*-BDL mice (*n* = 7, in each group) and were included for serial experiments.

### 5.3. Common Measurements between Two Sets of Mice

At the end point (week 6 after BDL) of the study, mouse was placed in a metabolic cage and 24-h urine samples were collected over 5 consecutive days, and the average of 5-day daily urine output (mL/100 g BW) was calculated. The supernatant of collected 3-day urine samples was used for measuring of urinary concentration of creatinine, and renal tubular epithelial damage markers (urinary levels of uKIM-1/creatinine (ng/mg u.cr) and urinary levels of uNGAL/creatinine ratio (ng/g tissue).

### 5.4. Experiments in the First Set of four Groups of Mice

#### 5.4.1. Systemic and Renal Hemodynamic Measurements

For the first set of 4 groups of mice (*n* = 7 in each group), cardiac output (CO), mean arterial pressure (MAP), and heart rate (HR) were measured. CO was normalized to body weight and represented as cardiac index (CI). Then, a midline incision was made in the abdomen. Whole kidney blood flow (RABF, mL/min.100 g) of the left and right kidney was measured. Renal vascular resistance (RVR) was calculated as the RABF divided by the MAP. Finally, tissues (liver, intestine, and kidney), mesenteric lymph nodes (MLN), stool and blood samples were collected for measurements of intestinal bacterial microbiota, intestine TNFα level, and plasma LPS-binding protein [LBP, BT marker, using ToxinSensor Chromogenic LAL Endotoxin Assay Kit (GenScript USA Inc. Piscataway, NJ, USA)].

#### 5.4.2. Measurements of Intestinal Permeability and Intestinal Inflammation

Evan Blue (EB)-permeated intestinal permeability methods were used to assess the degree of intestinal barrier dysfunction as previously described [55]. Additionally, intestinal permeability was reassessed by measurement of albumin content in the mice feces using ELISA kits (MyBioSource, Inc, San Diego, CA, USA). The data was normalized to the total weight of feces. For measurement of intestinal inflammation, the terminal ileum lumen was carefully cannulated, gently washed and embedded for staining with hematoxylene and eosin (H&E). Injury was classified using a semiquantitative grading system as shown in Appendix A.

#### 5.4.3. Measurements of Tissue Profiles

Proteins/*mRNAs* expressions of hepatic, intestine, renal inflammatory, oxidative stress, antioxidant, intestinal and renal injury as well as barrier markers were measured with appropriate antibodies/primers (Table 2). Tissue levels of SIRT1 activities were measured with SIRT1 fluorometric Kit (Abcam, relative fluorescence unit, RFU). Hepatic and renal collagen deposition and renal tubulointerstitial injury were measured with Sirius red and periodic acid-Schiff (PAS) staining. The average of the results of samples from each mouse were included for comparison.

#### 5.4.4. 16S rRNA Gene Sequencing Analysis for Intestinal Bacterial Microbiome

DNA extraction from fecal samples was conducted using a QIAamp Fast DNA Stool Mini Kit (Qiagen, Valencia, CA, USA) following the manufacturer’s protocols [18,20]. To obtain further insights into the pathogenic effects of genetic deletion of intestinal SIRT1 on intestinal bacterial microbiome, we used metagenomic sequencing of the 16S rRNA gene. The library for 16S rDNA amplicon sequencing was constructed based on the PCR-amplified V3–V4 variable regions. Alpha diversity (Shannon diversity, Faiths PD index, evenness index, and observed OTU) was calculated. The UniFrac principal coordinate analysis (PCoA), which evaluates phylogenetic similarities between microbial communities, was calculated for the beta-diversity.

#### 5.4.5. Direct Correlation between Intestinal Dysbiosis and Renal Dysfunction of Cirrhotic Mice

Correlation network between the abundance of significant bacteria and expression levels of intestinal and renal pathogenic markers were visualized by generating two interactive networks specific to the WT (WT-sham and WT-BDL mice) and KO (SIRT1*^IEC KO^* sham and SIRT1^*IEC KO*^ BDL mice) groups and analyzed using Cytoscape and CoNet. [56,57]. Additionally, the ratio of autochthonous (inhabiting a place or region from earliest time, *Lachnospiraceae* + *Ruminococcaceae* + *Veillonellaceae* + *Clostridialies* XIV) to non-autochthonous taxa (*Bacteroidaceae* + *Enterobacteriaceae*) was calculated as the cirrhosis dysbiosis ratio (CDR). Notably, the abovementioned autochthonous taxa on the numerator of CDR are bacteria that can reduce colonic inflammation and nourish colonocytes, compete with pathogenic bacteria for nutrients, avoid intestinal barrier dysfunction and reduce bacterial translocation [31,32]. It had been reported that low CDR is associated with endotoxemia, infection, death and organ failures within 30 days in cirrhotic patients [31,32]. In our study, the CDR of each mouse was calculated and correlated with the intestinal SIRT1 activity, the RVR data and the GFR data of individual group.

### 5.5. Experiments in the Second set of Mice

In in situ renal perfusion study, the dose-response of MAP, GFR and RVR to the incremental doses of TNFα (0.1, 0.3, and 0.5 ng/g/min) were measured in the second set of 4 groups of mice (*n* = 7 in each group). Two consecutive 30-min urine (basal period) and corresponding blood were collected for measuring plasma and urine concentrations of inulin, PAH and hematocrit. Then, the value for inulin clearance was considered as glomerular filtration rate (GFR) with formula of U_inulin_/P_inulin_*UV (U_inulin_/P_inulin_: urine and plasma levels of inulin, UV: urine volume in the given time period), and the value for PAH clearance was considered as renal plasma flow with formula of U_PAH_/P_PAH_*UV (U_PAH_/P_PAH_: urine and plasma levels of PAH, UV: urine volume in the given time period). Renal blood flow (RBF) was calculated from renal plasma flow and hematocrit values. Renal vascular resistance (RVR) was calculated by dividing the value of MAP with the value of RBF. The mean values obtained during the first two control collection periods were considered as “basal values” while the mean of the values collected during the two TNFα infusion periods was named as the “treatment value”. The differences in the values between the basal and the treatment periods were considered as the responses to TNFα treatment.

### 5.6. Statistical Analysis

Data were expressed as means ± S.D. Statistical significance for each group was determined using unpaired Student’s t test, one-way ANOVA, Newman−Keuls test, Mann−Whitney U-tests, or Wilcoxon signed rank test. Correlation analysis between the representative microbial genera, intestinal, and renal injury markers were analyzed with Pearson’s correlation coefficient and significant associations with *p* < 0.05 and r > 0.5 are shown.

## Figures and Tables

**Figure 1 ijms-22-01233-f001:**
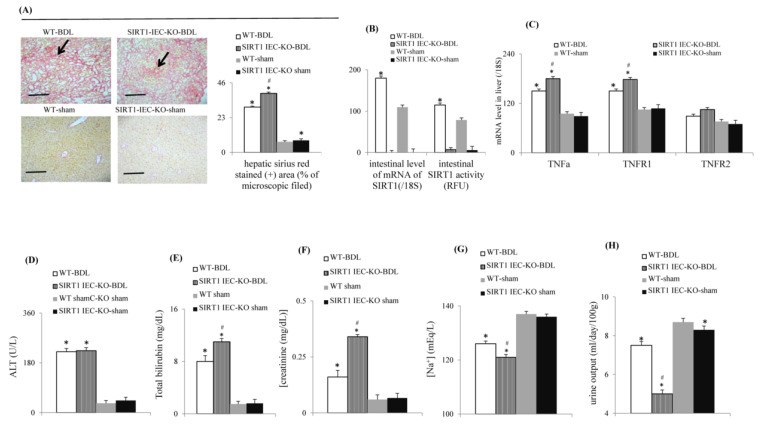
Intestinal SIRT1 deficiency aggravates hepatic fibrosis and renal dysfunction in bile duct ligated (BDL) cirrhotic mice with ascites. (**A**) Sirius red staining (+) (arrowhead) of mice liver section (20×, the scale bar is 100 μm); (**B**) intestinal SIRT1 levels; (**C**) hepatic levels of TNFα, TNFR1 and TNFR2 genes; (**D**) serum levels of ALT, (**E**) total bilirubin, (**F**) creatinine and (**G**) sodium; (**H**) urine output among groups. * vs. WT-sham mice; # vs. WT-BDL mice.

**Figure 2 ijms-22-01233-f002:**
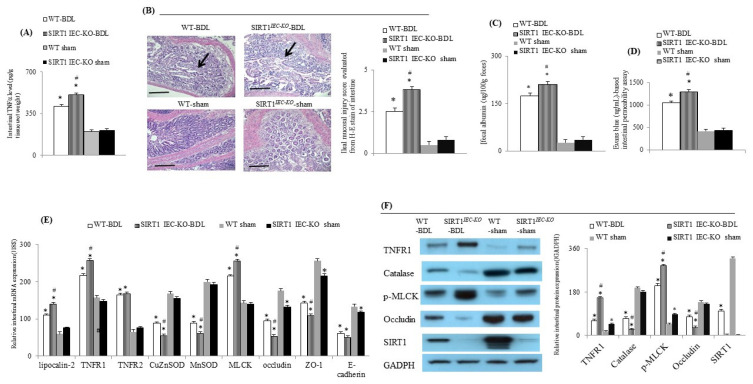
Intestinal SIRT1deficiency aggravates intestinal mucosal injury and barrier dysfunction in BDL-cirrhotic mice. (**A**) Intestinal TNFα level, (**B**) representative images (20×, the scale bar is 100 μm) and bar graph of H-E staining for the severity of ileal mucosal injury (arrowhead), (**C**) degree of fecal albumin loss, (**D**) degree of Evans blue-based intestinal barrier dysfunction, (**E**) relative mRNA and (**F**) protein expression of various TNFα-related signals among groups. * vs. WT-sham mice; # vs. WT-BDL mice.

**Figure 3 ijms-22-01233-f003:**
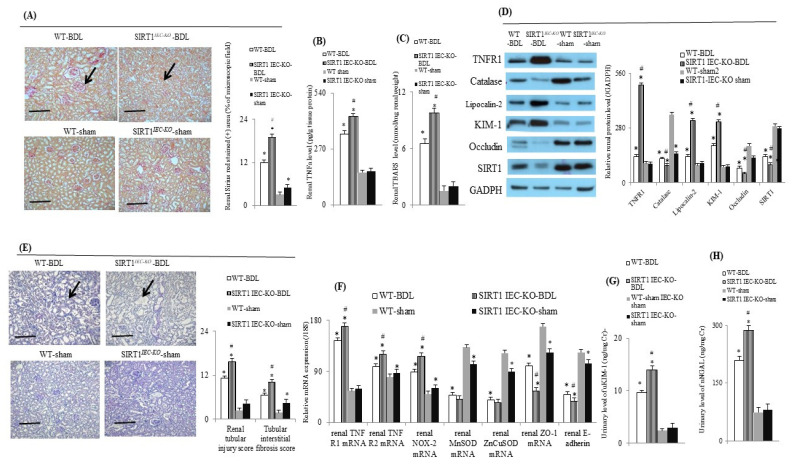
Intestinal SIRT1 deficiency increased renal TNFα-related inflammatory and oxidative stress-related markers. (**A**) Representative image and bar graphs of Sirius red staining (+) (arrowhead) of mice renal section (20×, the scale bar is 100 μm); (**B**) renal TNFα and (**C**) TBARS levels, (**D**) representative images and bar graphs of various renal TNFα-related beneficial and pathogenic proteins and (**F**) genes; (**E**) representative H-E staining image and bar graphs of mice renal section (20×, the scale bar is 100 μm) for the severity of renal injury (arrowhead); urinary levels of (**G**) KIM-1 and (**H**) NGAL among groups. TBARS: thiobarbituric acid-reacting substances. * vs. WT-sham mice; # vs. WT-BDL mice.

**Figure 4 ijms-22-01233-f004:**
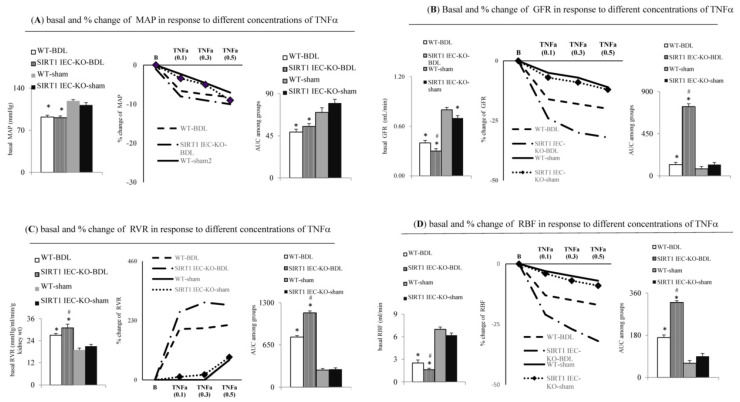
Effect of intestinal SIRT1 deficiency on systemic and renal vascular response to cumulative doses of TNFα cirrhotic ascitic mice. Graphs and area under curve (AUC) of concentration-response curve of TNFα induced a decrease in (**A**) the MAP, (**B**) the GFR; (**C**) the increase in the RVR and (**D**) the RBF during in situ renal perfusions among groups. * vs. WT-sham mice; # vs. WT-BDL mice.

**Figure 5 ijms-22-01233-f005:**
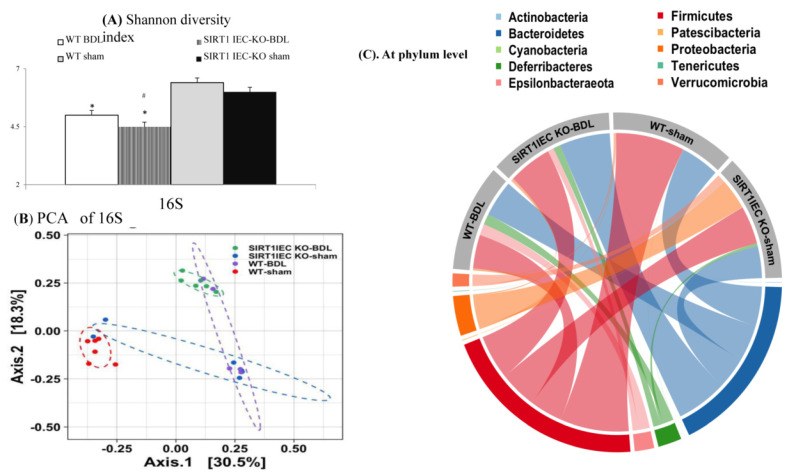
Genetic deletion of intestinal SIRT1 decreases the diversity of bacterial microbiota. (**A**) Simpson’s index for bacteria microbiota, (**B**) PCA plot among group, (**C**) chord diagram shows the parallel comparison of the overall abundance of intestinal microbiota of four groups. * vs. WT-sham mice; # vs. WT-BDL mice.

**Figure 6 ijms-22-01233-f006:**
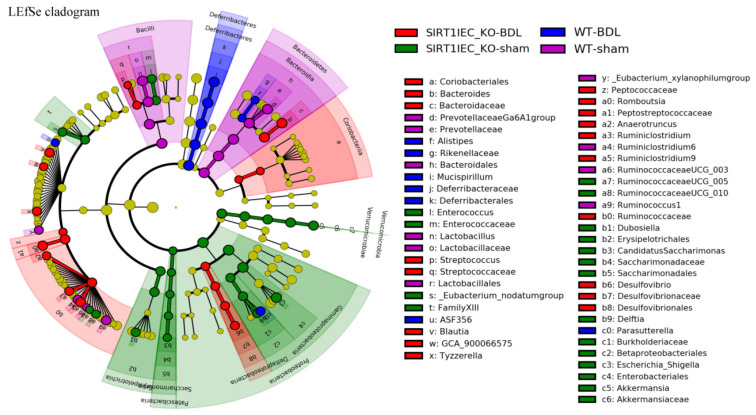
Intestinal SIRT1 deficiency decreases the diversity of bacterial microbiota. Intestinal LEfSe cladogram representing bacteria taxon with statistical significance and biological relevance fecal bacteria in WT BDL (blue), SIRT1*^IEC KO^*-BDL group (red), WT-sham (purple), SIRT1*^IEC KO^*-sham (green). Rings from the inside out represented taxonomic levels from phylum, family to genus levels. Sizes of circles indicate the relative abundance of the taxa.

**Figure 7 ijms-22-01233-f007:**
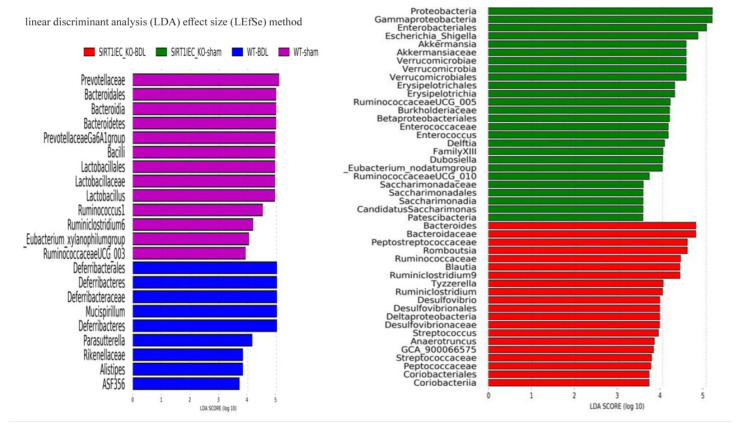
Intestinal SIRT1 deficiency aggravates cirrhosis-related intestinal bacterial dysbiosis. Overall distribution of statistical significance and biological relevance of intestinal bacteria among four groups [WT-BDL (blue), SIRT1*^IEC KO^*-BDL group (red), WT-Scheme 1. SIRT1*^IEC KO^*-sham (green)] that analysis by the linear discriminant analysis (LDA) effect size (LEfSe) method.

**Figure 8 ijms-22-01233-f008:**
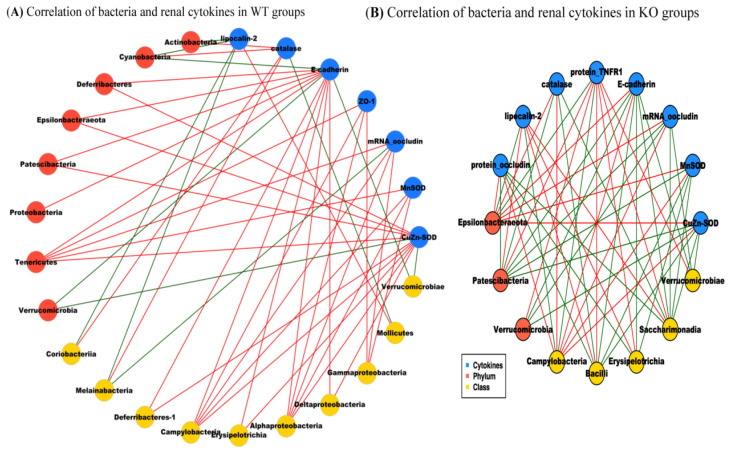
Crosstalk between intestinal and renal pathogenic markers in bile duct ligation (BDL) mice. (**A**,**B**) Significant correlation was noted between renal injury, renal inflammatory, anti-inflammatory, barrier markers and intestinal microbiota in WT groups (WT-BDL and WT-sham) and KO groups (SIRT1*^IEC-KO^*-BDL and SIRT1*^IEC-KO^*-sham). “Green” lines represent positive correlation between the abundance of pathogenic proteins and *mRNA* levels of corresponding pathogenic markers whereas “red” lines indicate the negative correlation between the abundance of the bacteria with protein and mRNA levels of corresponding pathogenic markers.

**Figure 9 ijms-22-01233-f009:**
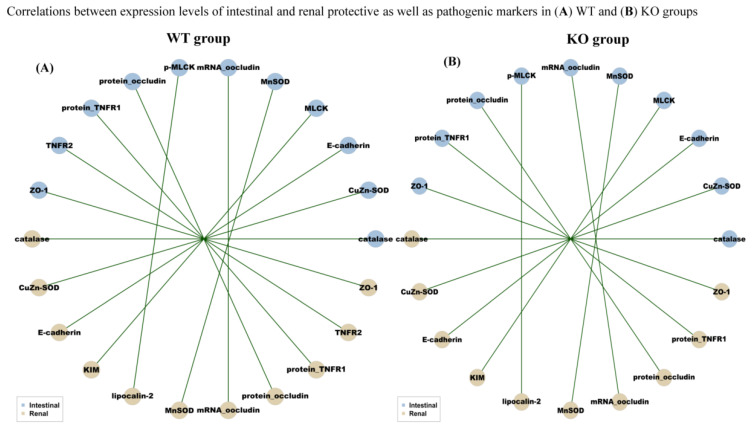
Crosstalk between intestinal/renal pathogenic markers and microbiota. Genetic deletion of intestinal SIRT1-related dysbiosis is associated with increased intestinal and renal pathogenic gene expressions. (**A**,**B**) Blue nodes: representative of intestinal or renal injury markers; brown nodes: differentially distributed bacteria genera in WT or KO groups. Green lines between nodes indicate positive relationships.

**Figure 10 ijms-22-01233-f010:**
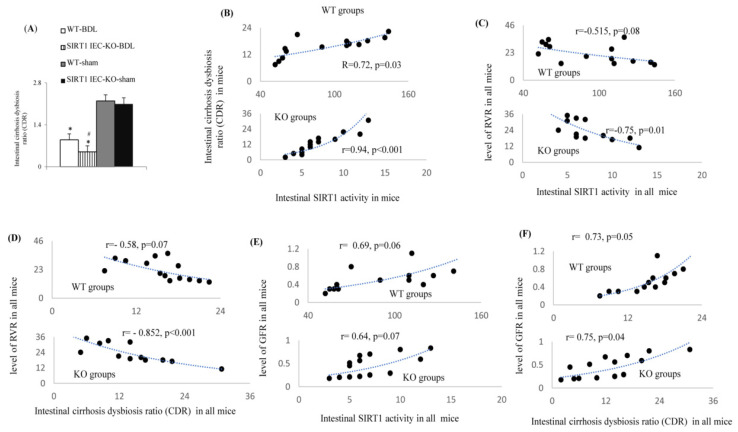
Relationship between various intestinal/renal pathogenic markers with cirrhosis discrimination ratio (CDR). (**A**) The data of CDR between groups; (**B**) correlation between intestinal SIRT1 activity and the CDR in WT and KO mice groups; * vs. WT-sham mice; # vs. WT-BDL mice; (**C**) correlation between intestinal SIRT1 activity and the RVR (renal vascular resistance) in WT and KO mice groups; (**D**) correlation between the CDR and the RVR in WT and KO mice groups; (**E**) correlation between intestinal SIRT1 activity and the GFR (glomerular filtration rate) in WT and KO mice groups; (**F**) correlation between the CDR and the GFR. WT mice groups include WT-BDL and WT-sham mice; KO mice groups include the SITR1*^IEC-KO^*-sham and SITR1*^IEC-KO^*-BDL mice. Mn/CuZn-SOD: superoxide dismutase; p-MLCK/MLCK: phosphorylated myosin light chain kinase; ZO-1: zonula occludens-1, TNFR1/TNFR2; tumor necrosis factor receptor 1 and 2; KIM; kidney injury marker.

**Table 1 ijms-22-01233-t001:** Hemodynamic parameters of mice in different groups.

	WT-BDL (*n* = 7)	SIRT1*^IEC-KO^*-BDL (*n* = 7)	WT-Sham (*n* = 7)	SIRT1*^IEC-KO^*-Sham (*n* = 7)
Body weight (gram)	27.9 ± 2.4 *	27.4 ± 3.2	30.8 ± 2.7	29.8 ± 4.6
Mean arterial blood pressure (MAP, mmHg)	92 ± 8 *	91 ± 10	119 ± 14	112 ± 7 ^#^
Heart rate (beats/min)	458 ± 44	461 ± 32	422 ± 51	430 ± 55
cardiac output (CO, mL/min)	7.4 ± 0.2 *	4.5 ± 0.2 ^#^	11.2 ± 0.3	10.9 ± 0.4 ^#^
cardiac index (CI, mL.min/100 g)	26.5 ± 1.7 *	16.7 ± 1.5 ^#^	36.3 ± 2.2	35.9 ± 2.9 ^#^
LPS binding protein (LBP) (ng/mL)	27.8 ± 1.2 *	34.7 ± 0.9	9.3 ± 0.6	12.4 ± 1.1
Culture % rate of MLN	43% *	71% ^#^	0	14%

* *p* < 0.05 vs. WT-sham group; ^#^
*p* < 0.05 vs. WT-BDL group; CI = CO/BW. MLN: mesenteric lymph nodes.

**Table 2 ijms-22-01233-t002:** Primer sequences used for various genes expression analysis by real-time qPCR.

Name of Gene	Sequence of Sense Primer (5′-3′)	Sequence of Anti-Sense Primer (3′-5′)
18S	GTAACCCGTTGAACCCCATT	CCATCCAATCGGTAGTAGCG
SIRT1	GCAACAGCATCTTGCCGAT	GTGCTACTGGTCTCACTT
TNFR1	TGACCCTCTCCTCTACGGA	CCATCCACCACAGCATACA
TNFR2	GACTGGCGAACTGCTT	AACTGGGTGCTGTGGTCAAT
Nox-2	GGGAACTGGGCTGTGAATGA	CAGTGCTGACCCAAGGAGTT
Nox-4	ACAGTCCTGGCTTACCTTCG	TTCTGGGATCCTCATTCTGG
CuZnSOD (superoxide dismutase)	GCGGTGAACCAGTTGTGTTGTC	CCTCTGGACCCGTTACACTGAC
MnSOD (superoxide dismutase)	ATGTTACAACTCAGGTCGCTCTTC	CCTCTCAACGACCTCCGATAGT
Catalase	CCGACCAGGGCATCAAAA	GAGGCCATAATCCGGATCTTC
MLCK	AAT GGT GTT GCT GGA GAT CGA GGT	CTCAAAGTTACCACCGCTGCTG
ZO-1	CGGGACTGTTGGTATTGGCTAGA	GGCCAGGGCCATAGTAAAGTTTG
Occludin	TCCTATAAATCCACGCCGGTTC	CTCAAAGTTACCACCGCTGCTG
E-cadherin	TCA ACG ATC CTG ACC AGC AGT TCG-	GGT GAA CCA TCA TCT GTG GCG ATG
KIM-1	TGGCACTGTGACATCCTCAGA	GCAACGGACATGCCAACATA
IL-18	CTTTGGAAGCCTGCTATAATCC	GGTCAAGAGGAAGTGATTTGGA
Lipocalin-2	TGGCCACTTGCACATTGTAG	ATGTCACCTCCATCCTGGTC

TNFR, receptor of tumor necrosis factor; MLCK: myosin light chain kinase, Interleukin-18 (marker of renal tubular injury).

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
