# Peer review of "Intestinal SIRT1 Deficiency-Related Intestinal Inflammation and Dysbiosis Aggravate TNFα-Mediated Renal Dysfunction in Cirrhotic Ascitic Mice"

_ijms, 2021, doi:10.3390/ijms22031233_

Round 1

Reviewer 1 Report

The manuscript entitled “Intestinal SIRT1 deficiency-related intestinal inflammation and dysbiosis aggravate TNFa-mediated renal dysfunction in cirrhotic ascitic mice” by Chou et al. is well planned, performed the experiments, and presented with sufficient data. The authors attempted to explore the impacts of intestinal SIRT1 deficiency and TNFα-mediated intestinal abnormalities on the development of cirrhosis-related renal dysfunction. The findings of this study will help to recover TNFa-mediated renal dysfunction in cirrhotic ascitic mice. I appreciate the authors for their effort to produce this manuscript. The introduction, results, and discussion parts are well written with enough background. The experimental protocols are clearly given in the material and methods section. Only minor corrections are required in this manuscript.

  1. Statical significance is missing in figure legends. what is mean by * and **?
  2. The conclusion of the study can be improved.

Author Response

Response to reviewer 1 comments:

Thanks for your very constructive opinion that improving our manuscript. We are appreciating for having this opportunity to revise our manuscript. The point-to-point response to your comments had been included as below

Comment 1: Statical significance is missing in figure legends. what is mean by * and **?

Response 1: Thanks for your comments. The statical significance was included in figure legends of Figure 1-5 and 10 in “revised” version (page 24-26).

Comment 2: The conclusion of the study can be improved.

Response 2: Thanks for your comment about conclusion. According to your and reviewer 2 opinion, some new reference 24-53-57 had been included in last paragraph of “discussion”. The “conclusion” had been rewritten to make it clear (page 13). 

Reviewer 2 Report

Title: Intestinal SIRT1 deficiency-related intestinal inflammation and dysbiosis aggravate TNFalpha-mediated renal dysfunction in cirrhotic ascitic mice

Authors: Yu-Te Chou, Tze-Tze Liu, Ueng-Cheng Yang, Chia-Chang Huang, Chih-Wei Liu, Shiang-Fen Huang, Tzu-Hao Li, Hsuan-Miao Liu, Ming-Wei Lin, Ying-Ying Yang, Tzung-Yan Lee, Yi-Hsiang Huang, Ming-Chih Hou, Han-Chieh Lin

Comments:

An overall well performed study to highlight the Intestinal SIRT1 deficiency-related intestinal inflammation and dysbiosis aggravate TNFalpha-mediated renal dysfunction in cirrhotic ascitic mice.

  1. A number of 14 authors is too much regarding the amount of data obtained in the presented study!
  2. The numbering of the authors in the authors list is not correct. It should be started with 1.
  3. Postal codes and mail addresses incomplete
  4. Please “cirrhotic ascites” should be entered as an additional keyword.
  5. Figures size (1A, 2B, 3A, 3E) is too small, labelling is missing and the quality is poor.
  6. Figures 5C/D, 6, 7A/B are barely visible. Simplify if possible, otherwise enlarge.
  7. The Western blot shown in Figure 2F, 3D, should still be shown blotting Sirt1.
  8. Abbreviations should be added extensively, for example missing:

CDR, RVR, GFR, HRS, AKI, DDS, IBD, CKD, RBF, MAP, PD, PCA, IEC, CO, CI, HR, RABF, EB.

  1. Table 1 comes after Table 2. Tables and text places should be renamed.
  2. Revise formatting:

- Table 2 is truncated

- Insert paragraphs at end of sections 2.1, 2.7, 4.4.3 and before Table 1

- Unify font size at the end of section 4.4.1 and within Table 1

- Unify References (2nd is italic, remove empty line behind 31, 48, 49, remove underlines)

  1. Check spelling:

- mice in section 2.2.

- Point at the end of the description of Figure 6

- patients in section 3.

- 4.2. dot after numbering is missing

- 4.3. common in heading capitalized

- 4.4.1. systemic in heading capitalized

- 4.5. dot at end of penultimate sentence missing

  1. Discussion: better and more detailed, e.g.

how the study results can be transferred to human medicine (are mice and humans comparable?) and

how the results can be implemented (what should the targeting look like?).

Resveratrol and Sirt1…….

  1. At the end of the discussion, the authors discuss that increased renal oxidative stress and inflammation are the main culprits for the deterioration of renal function in cirrhotic patients. Thus, agents that restore intestinal SIRT1 activity prevent TNF-mediated renal dysfunction in cirrhotic ascites mice. this is a very important point that needs further discussion.

One of the major natural activators of Sirt1 is resveratrol.

Please, the author should discuss more about the signaling pathways of anti-inflammatory and antitumor effects of resveratrol via Sirt1 in kidney and other tissues. Please add additional sentences and references: 

Li P et al., (2020). Resveratrol improves left ventricular remodeling in chronic kidney disease via Sirt1-mediated regulation of FoxO1 activity and MnSOD expression. Biofactors. 46(1):168-179.

Buhrmann C et al., (2020). Evidence that TNF-β suppresses osteoblast differentiation of mesenchymal stem cells and resveratrol reverses it through modulation of NF-kB, Sirt1 and Runx2. Cell Tissue Res. 381(1):83-98.

Zhang J et al., (2019). Catalpol alleviates adriamycin-induced nephropathy by activating the SIRT1 signalling pathway in vivo and in vitro. Br J Pharmacol. 176(23):4558-4573.

Buhrmann C et al., (2016). Sirt1 is required for resveratrol-mediated chemopreventive effects in colorectal cancer cells. Nutrients, 8: 2-21.     

Liu S et al., (2019). Resveratrol exerts dose-dependent anti-fibrotic or pro-fibrotic effects in kidneys: A potential risk to individuals with impaired kidney function. Phytomedicine. 57:223-235.

Buhrmann C et al., (2014). Sirt-1 is required for promoting chondrogenic differentiation of mesenchymal stem cells. J Biol Chem, 289: 22048-22062.

  1. Conclusion: as a separate paragraph after discussion would be nice.

Author Response

Response to reviewer 2 comments:

Thanks for your very constructive opinion that improving our manuscript. The point-to-point response had been included as below.

Comment 1: A number of 14 authors is too much regarding the amount of data obtained in the presented study!

Response 1: We had the review opinion in our lab meeting. Thanks for your comment about number of author, according to you suggestion we had removed one of the co-author (Chih-Wei Liu) from the co-author list to the acknowledge section according to the contribution to the work. The announcement for agreement for removal from the co-author had been included as below. We will ask editor office to help us to remove his name from the submission website. Thanks for giving us this opportunity to check the contribution of each author in the list.

Comment 2: The numbering of the authors in the authors list is not correct. It should be started with 1.

Response 2: Thanks for your suggestion about the number of affiliations. In “revised” version, we had began from number 1 (page 1).

Comment 3: Postal codes and mail addresses incomplete

Response 3: Thanks for your suggestion about the postal codes and mail addresses. In revised title, the postal codes and mail addresses were included in title page (page 1).

Comment 4: Please “cirrhotic ascites” should be entered as an additional keyword.

Response 4: Thanks for your suggestion, the “cirrhotic ascites” had been included as an additional keyword in “revised” version (page 3).

Comment 5: Figures size (1A, 2B, 3A, 3E) is too small, labelling is missing and the quality is poor.

Response 5: Thanks for your suggestion about the size, labelling and quality of (1A, 2B, 3A, 3E). In “revised” version, we had adjusted these points.

Comment 6: Figures 5C/D, 6, 7A/B are barely visible. Simplify if possible, otherwise enlarge.

Response 6: Thanks for your comment about the Figure 5C/D, 6, 7A/B. In revised version, we had adjusted the quality of images and enlarged the individual images. The new labeled of these images are Figure 6 (original Figure 5D), Figure 7 (original Figure 6A), Figure 8A-B (original Figure 6B-C), Figure 9A-B (original Figure 7A-B), Figure 10A-F 9 (original figure 7C-H). The original Figure 8 had changed as supplement Figure 5.

Comment 7: The Western blot shown in Figure 2F, 3D, should still be shown blotting Sirt1.

Response 7: According to your suggestion, the western blot of intestinal and renal Sirt 1 expression levels were included in Figure 2F and 3D.  

Comment 8: Abbreviations should be added extensively, for example missing: CDR, RVR, GFR, HRS, AKI, DDS, IBD, CKD, RBF, MAP, PD, PCA, IEC, CO, CI, HR, RABF, EB.

Response 8: According to your suggestion, the missing abbreviation had been included in abbreviation section (page 18&19).

Comment 9: Table 1 comes after Table 2. Tables and text places should be renamed.

Responses: Thanks for your instruction about the order of table 1 and 2. In revised version the table 1 had changes into table 2 whereas table 2 had changed into table 1. The tables and text were renamed and replaced accordingly.

Comment 9:  Revise formatting:

Question- Table 2 is truncated

Response: In “revised” version, the table 2 (new table 1) had been adjusted to avoid truncated.

Question- Insert paragraphs at end of sections 2.1, 2.7, 4.4.3 and before Table 1

Response: the changing of Table 1 into Table 2 had been revised in abovementioned paragraphs.

Question - Unify font size at the end of section 4.4.1 and within Table 1.

Response: Thanks for your kindly instruction for the references. We had carefully adjusted the font size at the end of section 5.4.1 (original 4.4.1) and within Table 1.

Question - Unify References (2nd is italic, remove empty line behind 31, 48, 49, remove underlines)

Response: Thanks for your kindly instruction for the references. We had carefully checked the style of references. All the volume of paper had been changed to italic. The under liens were removed.

Comment 12: Check spelling:

Question- mice in section 2.2.

Response: Thanks for your kindly instruction, the misspelling and wording of this section had been checked and corrected.

Question- Point at the end of the description of Figure 6

Response: Thanks for your kindly instruction, the missing point at the end of description of Figure 6 had been included in “revised” methods.

Thanks for your comments as the conclusion had been moved as separated paragraph with number of section 4. So, the methods had labeled as section 5 in “revised” methods.

- 5.2. dot after numbering is missing

Response to 5.2.: Thanks for your kindly instruction, the missing dot after numbering had been added in “revised” methods (page 14, paragraph 2, line 7).

- 5.3. common in heading capitalized

Response to 5.3.: Thanks for your kindly instruction, the “Common” had been capitalized in “revised” methods (page 14, paragraph 3, line 1).

- 5.4.1. systemic in heading capitalized.

Response to 5.4.1.: Thanks for your kindly instruction, the “Systemic” had been capitalized in “revised” methods (page 14, paragraph 4, line 2)..

- 5.5. dot at end of penultimate sentence missing.

Response to 5.5: Thanks for your kindly instruction, the missing dot at the end of penultimate sentence had been added in “revised” methods (page 16, paragraph 2, line 16).

Comments 13: Discussion: better and more detailed, e.g.how the study results can be transferred to human medicine (are mice and humans comparable?) and how the results can be implemented (what should the targeting look like?). Resveratrol and Sirt1…….; At the end of the discussion, the authors discuss that increased renal oxidative stress and inflammation are the main culprits for the deterioration of renal function in cirrhotic patients. Thus, agents that restore intestinal SIRT1 activity prevent TNF-mediated renal dysfunction in cirrhotic ascites mice. this is a very important point that needs further discussion.

Response 13: Thanks for your suggestion to discussion these important points. In “revised” discussion (page 14, paragraph 3), the aforementioned discussion had been included as “This study reenforced the concepts of increased TNFa-mediated renal oxidative stress and inflammation are the main culprits for the deterioration of renal function in cirrhotic patients. In cirrhotic ascitic mice, the well-expressed intestinal SIRT1 is crucial to prevent TNFa-mediated activation of systemic, intestinal and renal oxidative stress, inflammation and injured signals that contributing to the development of renal dysfunction. So, agents that restoration of intestinal SIRT1 are potential to improve the inflammation-derived TNFa-mediated renal dysfunction in advanced cirrhosis. So, it is noteworthy to explore the effects of the administration of highly intestinal-absorbed SIRT1 activator such as resveratrol on the co-existed TNFa-mediated intestinal and renal pathogenic changes as well as renal dysfunction in cirrhosis”.

Comments 14: One of the major natural activators of Sirt1 is resveratrol. Please, the author should discuss more about the signaling pathways of anti-inflammatory and antitumor effects of resveratrol via Sirt1 in kidney and other tissues. Please add additional sentences and references: 

Response 14: Thanks for your comments and detail lists of the references about the anti-inflammatory and antitumor effects of resveratrol via Sirt1 in kidney and other tissues. The six references (ref. 24,53-57) had been included in lst paragraph of discussion in revised version as below (page 14, paragraph 2). Thanks for your comments to improve our discussion.

 Resveratrol has rapid metabolism and low bioavailability. After resveratrol was ingested 77–80% of it was absorbed in the intestine and 49–60% of this is excreted in the urine. After ingestion resveratrol also reaches the intestine via the hepatic portal system. Decreased intestinal SIRT1 had been associated with intestinal and renal inflammation [18,19,24]. Anti-TNFa-related anti- inflammatory effects of SIRT1 activator resveratrol had been reported in chondrogenic mesenchymal stem and tuman colon rectal cancer cell system [53-55Constanze, B,; Popper, B,; Aggarwal, B,B,; Shakibaei,  M. Evidence that TNF-β suppresses osteoblast differentiation of mesenchymal stem cells and resveratrol reverses it through modulation of NF-kB, Sirt1 and Runx2. Cell. Tissue. Res. 2020, 381(1),83-98 (ref. 53); Buhrmann, C,; Busch, F,; Shayan, P,; Shakibaei, M. Sirt-1 is required for promoting chondrogenic differentiation of mesenchymal stem cells. J. Biol. Chem. 2014, 289, 22048-22062 (ref. 54); Buhrmann, C,; Shayan, P,; Popper,  B,; Goel, A,; Shakibaei,  M. Sirt1 is required for resveratrol-mediated chemopreventive effects in colorectal cancer cells. Nutrients 2016,8,2-21(ref. 55)].  In experimental models of acute and chronic kidney injury, chronic systemic SIRT1 activation with resveratrol significantly improves adriamycin‐induced, subtotal nephrectomy-induced and unilateral ureteral obstruction-induced renal dysfunction through the anti-inflammatory effects [24,56-57 Li,  P,; Song, X,; Zhang, D,; Guo, N,; Wu, C,; Chen, K,; Liu,  Y,; Yuan, L,; Chen, X,; Huang, X. Resveratrol improves left ventricular remodeling in chronic kidney disease via Sirt1-mediated regulation of FoxO1 activity and MnSOD expression. Biofactors. 2020, 46(1),168-179(ref.56);Zhang, J,; Bi,  R,; Meng,  Q,; Wang, C,; Huo, X,; Liu,  Z,; Chong, Wang, C,; Sun, P,; Sun, H,; Ma, X. Catalpol alleviates adriamycin-induced nephropathy by activating the SIRT1 signalling pathway in vivo and in vitro. Br. J. Pharmacol. 2019,176(23),4558-4573 (ref. 57)]. Through SIRT1 activation, resveratrol treatment attenuated the intestinal inflammation and renal dysfunction in non-cirrhosis model [17,20,21,24,53-57]. This study reenforced the concepts of increased TNFa-mediated renal oxidative stress and inflammation are the main culprits for the deterioration of renal function in cirrhotic patients. In cirrhotic ascitic mice, the well-expressed intestinal SIRT1 is crucial to prevent TNFa-mediated activation of systemic, intestinal and renal oxidative stress, inflammation and injured signals that contributing to the development of renal dysfunction. So, agents that restoration of intestinal SIRT1 are potential to improve the inflammation-derived TNFa-mediated renal dysfunction in advanced cirrhosis. So, it is noteworthy to explore the effects of the administration of highly intestinal-absorbed SIRT1 activator such as resveratrol on the co-existed TNFa-mediated intestinal and renal pathogenic changes as well as renal dysfunction in cirrhosis.

Comment 15: Conclusion: as a separate paragraph after discussion would be nice.

Response 15: According to your suggestion, the conclusion had been included as separate paragraph after discussion in “revised” version (page 14, paragraph 3).

Round 2

Reviewer 2 Report

The authors have satisfactorily addressed the concerns raised in the original version. The revised version is significantly improved. No further concerns.